



# Hourly mass and snow energy balance measurements from Mammoth Mountain, CA USA, 2011-2017

Edward H. Bair[1], Robert E. Davis[2], and Jeff Dozier[3]

[1]Earth Research Institute, University of California, Santa Barbara, CA 93106-3060, USA
[2]Cold Regions Research and Engineering Laboratory, Hanover, NH 03755, USA
[3]Bren School of Environmental Science & Management, University of California, Santa Barbara, CA 93106-5131, USA

*Correspondence to*: Edward H. Bair (nbair@eri.ucsb.edu)

**Abstract:**

The mass and energy balance of the snowpack govern its evolution. Direct measurement of these fluxes is essential for modeling
the snowpack, yet there are few sites where all the relevant measurements are taken. Mammoth Mountain CA USA is home to the
Cold Regions Research and Engineering Laboratory and University of California – Santa Barbara Energy Site (CUES), one of five
energy balance monitoring sites in the Western US. There is a ski patrol study site on Mammoth Mountain, called the Sesame
Street Snow Study Plot, with automated snow and meteorological instruments where new snow is hand weighed to measure its
water content. For this dataset, we present a clean and continuous hourly record of selected measurements from both sites covering
the 2011-2017 water years which can be used to run a variety of snow models. The 2011-2017 period was marked by exceptional
variability in precipitation, even for an area that has high year-to-year variability. The driest year on record, and one of the wettest
years, occurred during this time period, making it ideal for studying climatic extremes. This dataset complements a previously
published dataset from CUES containing a smaller subset of daily measurements. In addition to the hand weighed SWE, novel
measurements include hourly broadband snow albedo corrected for terrain and other measurement biases. This dataset is available
with Digital Object Identifier: 10.21424/R4159Q.

## 1  Introduction

The mass and energy balance of the snowpack govern its evolution. Direct measurement of the variables that comprise these
balances is critical to our understanding of the snowpack. Yet, direct measurement of all necessary variables is rare, especially at
high-altitude sites. Additionally, there are some variables, such as the broadband snow albedo, which require tedious and nontrivial
adjustments that are only possible by those who are intimately familiar with the measurements and the site where they were taken.
In the Western US, there are five such sites where the full energy balance is monitored (Bales et al., 2006). One of these sites is
the Cold Regions Research and Engineering Laboratory and University of California – Santa Barbara Energy Site (CUES). CUES
has many unique features and, over the decades, has been home to numerous snow hydrology and snow avalanche studies. Bair et
al. (2015) describe the history of CUES, summarize current measurements, and provide three case studies using these
measurements. We provide here an expansive dataset, with hourly measurements of all the variables required to model the snow
mass and energy balance.

## 2  Study areas

CUES (37.643 N, 119.029 W) is located at 2940 m, midway up Mammoth Mountain, CA USA (Figure 1). The Sesame Snow
Study Plot (37.650 N, 119.042 W, elevation 2743 m) is located just above the Main Lodge at Mammoth Mountain (Figure 1).




Mammoth Mountain is a silica dome cluster (Hildreth, 2004) in the central Sierra Nevada. It is an active volcano with eruptions as recent as 300 to 700 years ago, based on evidence from radiocarbon dated samples of charred wood (Bailey, 1989). There are several active fumaroles on Mammoth Mountain, and its surface is covered by volcanic deposits. As it relates to snow hydrology and albedo degradation, large portions of Mammoth Mountain are coved by tephra or pumice. In fact, prior to being named

5   Mammoth Mountain, it was rumoured to be called *Pumice Mountain* by local inhabitants. Strong winds often blow pumice onto the snow surface and significantly degrades its albedo (Sterle et al., 2013).

One of the largest ski areas in North America, Mammoth Mountain currently has 28 ski lifts including a gondola that operates nearly year-round, making CUES highly accessible relative to other high altitude scientific research sites. The CUES site itself is located on a small plateau. Vegetation consists of loosely spaced trees, mostly Whitebark and Lodgepole Pine, with some shrubs

10  in the understory that are usually buried after the first significant snowfall. The loosely spaced trees and its topography give CUES exposure to most of the sky, but also expose the site to strong winds that make accurate measurement of precipitation with typical weighing gauges impossible. Instead, snow depth sensors and snow pillows are used. For precipitation, we use data from the nearby Sesame Street Snow Study Plot (Bair, 2013). Unlike CUES, the Sesame Plot is located in a small opening in a Whitebark Pine forest. The understory here also consists of small shrubs. Ground cover is also predominately tephra.

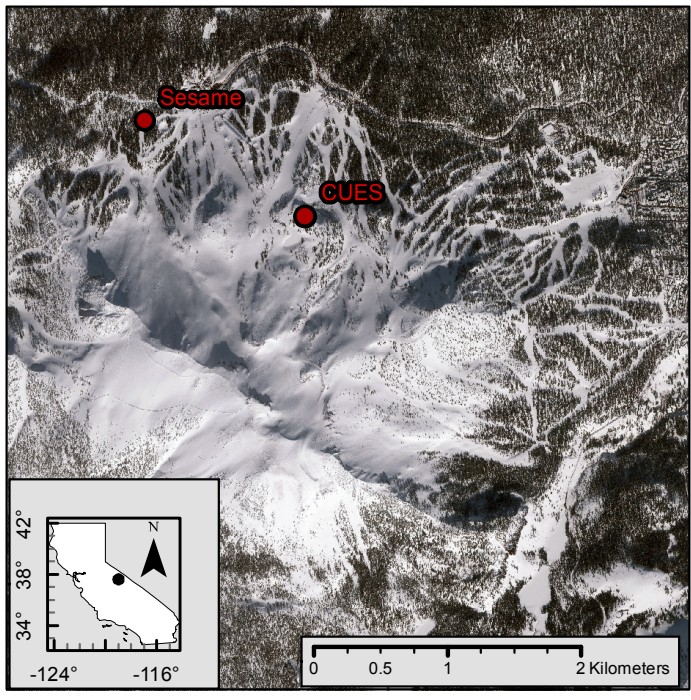

**Figure 1. Satellite imagery of Mammoth Mountain on 2017-02-04 showing the Sesame Snow Study Plot and CUES. The satellite image is from DigitalGlobe Worldview-3.**

## 2.1  Sesame Snow Study Plot

Hand weighed snowfall measurements every 24 hr at the Sesame Street Snow Study Plot (Figure 2), hereafter called Sesame, use

20  a white wooden board that is cleared daily, each time enough snow to accurately weigh falls (a few cm). At least two cores are made and the average is taken. We provide all the manual Sesame Snow Study Plot measurements (Table 1) for days with




precipitation, based on the morning daily weather observations, posted on as the "Storm Summaries" on http://patrol.mammothmountain.com. Rounding to the nearest hour, these measurements are almost always recorded at 7 am.

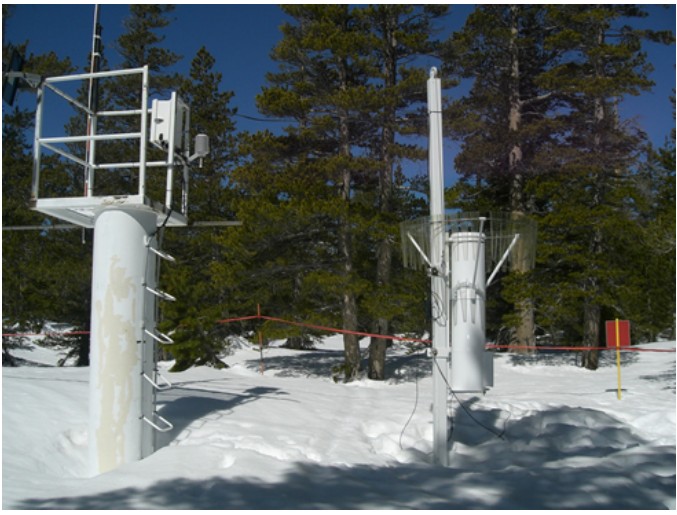

**Figure 2. Sesame snow study site in April 2006. On the left is the tower where total snow depth is measured using a boom. On the right**
**is a weighing gauge, manually raised and lowered about 1 m above the snow surface.**

Sesame also has two automated precipitation gauges, a MetOne 385 Rain Gauge tipping bucket and a Sutron Total Precipitation weighing gauge, both with one minute readings; but the measurements from these gauges had large gaps from times when they were not working, making them unsuitable for a continuous hourly dataset. Sesame has two ultrasonic Judd snow depth sensors also, one for the 24-hr board and one to measure the total snow depth. We provide the total depth here, as the automated 24-hr

board snow depth measurements are more useful for operational purposes and can be difficult to interpret, for instance when the board is being cleared. A snow pillow was installed by the California Department of Water Resources (DWR) and the first author in 2013 Oct. It has continuous measurements since then except for a period starting sometime around 2017 Feb when the pressure transducer failed, presumably because of the exceptional weight of the snowpack. The transducer was replaced in 2017 Jul. Because the snow pillow measurements only started in the fall of 2013, they are not part of this dataset, but are available upon request.

Likewise, a Lufft WS600-UMB Smart Weather Sensor was installed at Sesame in 2012 Jan. This sensor is equipped with a Doppler radar and uses a mass fallspeed relationship to estimate precipitation rates. For liquid precipitation this method works quite well (Löffler-Mang et al., 1999), but for solid precipitation, this method requires assumptions about the snowflake mass and other properties, giving inaccurate snowfall rate (Matrosov, 2007). We have noted inaccurate results when comparing the WS600 estimates to the hand weighed estimates at Sesame. Additionally, the WS600 shows many false positives for precipitation when

no precipitation occurs. The WS600 is useful at Sesame because it contains a sonic anemometer that measures wind speed, and this wind speed measurement can be used to correct for undercatch in the precipitation gauges (Goodison et al., 1998). Comparisons of hand weighed and automated measurements at Sesame show an undercatch of 9% on average.

Since Sesame is an operational ski area site, precipitation measurements during the summer are not reliable. We suggest this is not a significant problem given our focus on snow mass and energy balance measurements. Also, summer rainfall in the Sierra Nevada

is a small fraction of total precipitation. In the Sierra Nevada, October through May precipitation accounts for about 95% of annual precipitation (NOAA National Climatic Data Center, 2017). Nonetheless, we emphasize to the reader that reliable precipitation



estimates from Sesame only comprise periods starting with the season's first snowfall, usually in October, prior to the opening of the ski area, until the ski area closes, usually between 31 May and 4 Jul.

### 2.2 CUES

As with Sesame, mass balance measurements are focused on the snow accumulation and ablation season at CUES (Figure 3). The
site operates year-round, but as discussed earlier, there are no reliable precipitation gauge measurements, as the site is too windy for reasonable catch efficiencies. As with Sesame, a Lufft WS600-UMB was installed at CUES in 2011 Oct, but given the inaccuracies and our experience with this sensor described in Section 2.1 and since its measurements do not cover the entire study period, we have not included its measurements in this dataset.

Snowfall is measured most consistently at CUES using ultrasonic snow depth sensors (Table 2). Currently, there are three sensors
at the site. For this study, a Campbell SR50A Sonic Ranging sensor located underneath the RM Young 5103 Anemometer was used (Table 2).

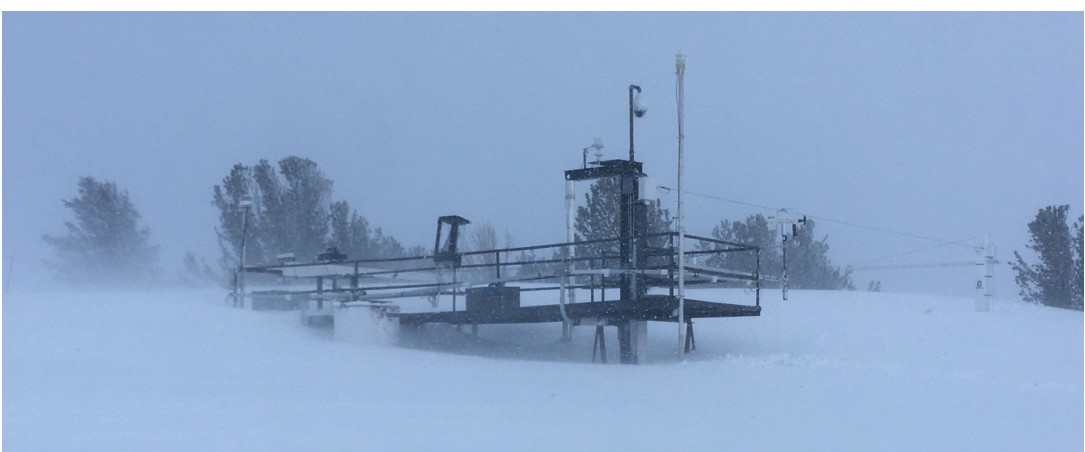

**Figure 3. A nearly buried CUES platform in 2017 Feb. The RM Young 5103 is the white anemometer with a propeller. The SR50A is the downlooking silver sensor below. The snow depth has reached the CUES platform, 6 m above the ground, several times since 1987, but**
**never completely buried it.**

The other two snow depth sensors at CUES are ultrasonic Judd depth sensors, the same models used at Sesame. The SR50A was used for this study because it outputs a quality flag for each depth indicating whether the target is solid or not, as ultrasonic depth sensors show considerable noise. In theory, this quality flag simplifies the filtering process. In reality, the snow depth measurements were still quite noisy and required substantial manual cleaning (Section 3.2.1).

A snow pillow was installed by the DWR in 2012 Sep and CUES has hosted experimental fluidless and other snow pillows in the past. Measurements from the DWR pillow and fluidless pillows are not included here because they do not span the entire length of the dataset.

## 3 Datasets

### 3.1 Energy balance measurements

The energy balance measurements come entirely from CUES. Some of these measurements, such as the incoming radiation, are available going back to 1992 at www.snow.ucsb.edu. Uplooking broadband solar radiation, both diffuse and direct, is provided in



this dataset by the Sunshine Pyranometer SPN-1. Incoming longwave radiation is measured via an Eppley Precision Infrared Radiometer (PIR). The broadband and near infrared (nIR) snow albedo are measured using uplooking and downlooking radiometer pairs, with the downlooking radiometers on a fixed boom prior to 2016 Sep and on an adjustable boom thereafter. Air temperature and relative humidity are measured simultaneously using a radiation shielded Campbell HMP 45C. Atmospheric pressure is

measured using a Campbell Scientific CS100. Wind speed and direction are measured using three anemometers: an RM Young 81000 Ultrasonic Anemometer, a Lufft WS600 UMB, and an RM Young 5103. The RM Young 81000 measures the 3-D wind vectors at a high sampling rate (10 Hz) for application of the eddy covariance method (e.g. Reba et al., 2009) to estimate sensible and latent heat fluxes. Because of the high sampling rate, the time series of the 3-D wind components is large, 52 GB. Processing this massive time series over the entire study period is impractical and therefore beyond the scope of this study. Because of the

size of those data, unlike almost all the other raw measurements from CUES, they are not available at [www.snow.ucsb.edu](www.snow.ucsb.edu), but are available upon request.

### 3.2   Data filtering and processing

#### 3.2.1   Mass balance

The manual Sesame measurements were checked visually for errors and edited to be consistent from year-to-year and adjusted to

conform to operational standards (American Avalanche Association, 2016), but no major adjustments were made. Minor adjustments included altering the table to show "Trace" amounts of new snow in a consistent way and some formatting changes. For example, null or blank values that were not applicable such as new snow density on days with rain only were converted to "NaN" (not a number).

The snow depth measurements from the ultrasonic depth sensors at the Sesame Snow Study Site and at CUES required extensive

filtering and interpolation, which is common. Snow depth measurements were aggregated from their native temporal resolution to 1 hr and retrieved using a database where both the Sesame and CUES measurements are stored. The database aggregation produces a forward looking average meaning that the average for say 12:00 contains the average of all measurements from 12:00 to 12:59. Note this is different than how most dataloggers average, which is a reverse looking average, e.g. for this case the 12:00 average would contain all measurements from 11:00 to 11:59. The two methods can be easily converted between by adding or subtracting

1 hr from the DateTime field.

For the Sesame site, the raw depth measurements are taken every minute. At CUES, the depth measurements are taken every 15 min from 2010 Oct until 2012 Sep. From 2012 Sep onwards, the depth measurements are taken every 5 min. Ultrasonic snow depth measurements suffer from both drops and spikes, but spikes are more prevalent, especially at CUES where blowing snow can reflect the sound, thereby causing the snowpack to appear to be at or near the sensor height. Thus, the aggregation used the

minimum depth over each hr to limit spikes. The aggregated data were then plotted and inspected visually. First, snow free periods were identified manually based on visual inspection of the measurements and ancillary knowledge of the snow accumulation season, e.g. Sesame manual measurements. These manually identified periods were set to zero depth. Other large spikes over extended periods of time, such as during sensor maintenance, were set to a missing value. Missing values were then interpolated using a shape-preserving piecewise cubic spline. The interpolated data were then smoothed using a smoothing spline to reduce

high frequency noise. This method tended to produce very small values (<< 0.1 cm) rather than zeros at times. These small values were set to zero. Measurement source is listed for all measurements in the table.



### 3.2.2 Energy balance

Air temperature, relative humidity, and air pressure were the simplest variables to process. Hourly averages were queried from the database containing measurements at 1 min temporal resolution. Visual examination of plots of these data revealed a few out-of-bounds measurements that were set to missing values. As with all the measurements, there were gaps of an hr to a few weeks, mostly in the summer, when instruments were removed for calibration or had failed. We arbitrarily chose a gap threshold of 12 hr, then used two approaches for gap-filling. For gaps below the gap threshold, a spline interpolation was performed. For gaps at or above the threshold, measurements were filled using climatological averages from the same day and time of year from all other years. As with the automated mass balance measurements, measurement source is listed in the table.

Wind speed and direction were queried from the database as average hourly values from 1 min samples. The Yamartino (1984) approach was used to average wind directions, consistent with the 1 min averaging that takes places on the dataloggers with the raw measurements. CUES has three different wind sensors, but only the RM Young 5103 and the Lufft WS600-UMB provide reliable 1 min averages (Table 2). The RM Young 5103 and the WS600 are about 2 m apart in height, with the 5103 being at the platform height of about 6 m above the bare ground. The WS600 is mounted higher, about 3 m above the platform surface. Both of the wind sensors had periods of missing data or high readings, e.g. > 60 m/sec for an hourly average. Thus, the RM Young 5103 values were preferentially used, with the WS600 measurements used to fill gaps. After this step, there were some small remaining gaps that were interpolated with a spline or filled with climatology using the same 12 hr gap threshold. The measurement instrument and type of processing (i.e. measured, interpolated, or climatology) is recorded in the data table.

The direct $B_\downarrow$ and diffuse $D_\downarrow$ broadband radiation were queried from the database as average hourly values from 1 min samples. Direct broadband radiation values where transmittance $T$ was above 0.95 were adjusted such that $T = 0.95$ on the assumption that the absolute measurement errors for direct radiation are greater than for diffuse radiation.

$$T = \frac{B_\downarrow + D_\downarrow}{\cos\theta_0 \left(\frac{S}{R_v^2}\right)} \tag{1}$$

where $\theta_0$ is the solar zenith angle, $S = 1367\,\text{W/m}^2$ is the solar constant and $R_v$ is the radius vector. These values can all be determined using location, date/time, and Ephemeris estimates, e.g. from the National Oceanic and Atmospheric Administration's Solar Calculator (https://www.esrl.noaa.gov/gmd/grad/solcalc/).

Periods of time with missing uplooking radiation were filled in with radiation from the nearby Dana Meadows, California Data Exchange Center (California Department of Water Resources, 2017) code DAN, at 37.897 N, 119.257 W at 2987 m. Comparisons of total solar radiation from CUES and DAN show decent though not excellent agreement, $r^2 = 0.85$, RMSE = 133 W m$^{-2}$. When DAN was the incoming solar radiation source, direct and diffuse components were estimated using an empirical method (Erbs et al., 1982) with a high altitude modification (Olyphant, 1984). Finally, there were a few periods left when neither CUES nor DAN had working uplooking broadband radiometers. These periods were multiple days in length so climatology was used to estimate the incoming solar radiation. After plotting the gap-free dataset, there were some obvious spikes at night or around sunrise/sunset, therefore all uplooking solar measurements are set to zero when $\cos\theta_0 \leq 0$; that is when the sun was below the horizon. This will zero out some of the low energy diffuse radiation at these times, but they are so small as to be insignificant to the energy balance.

The uplooking longwave radiation was relatively error free and required almost no filtering. There were gaps in the measurement record however that were filled using an empirical approach (Marks and Dozier, 1979) based on air temperature, and relative humidity when these ancillary measurements were available. We note that this approach is optimal for clear skies and produces



low biased values, but it was only used for < 1% of the uplooking longwave measurements. If these ancillary measurements were not available, climatology was used.

The snow albedo estimates were by far the most complicated to process. For snow albedo measurement at CUES, there are four downlooking radiometers: a clear and nIR Eppley PSP on a fixed boom, and a clear and nIR Eppley PSP on an adjustable boom. In theory, these measurements can be used in conjunction with the uplooking radiometers to measure snow albedo. In practice, there are many biases and steps that need to be taken to obtain accurate measurements. Potential sources of error include a sloped snow surface such that the level uplooking radiometers are not receiving the same amount of solar radiation as the snow; shadows cast by trees or other objects that can affect the uplooking and downlooking radiometers at different times; non-snow objects in the radiometers' field of view; an inability for the downlooking radiometers to distinguish diffuse radiation from the sky from that from the snow; direct solar radiation reaching the downlooking radiometers at high solar zenith angles; and imperfect cosine response and other instrument biases in the radiometers (Wilcox and Myers, 2008), especially at the higher solar zenith angles. To address the issue of non-snow objects in the downlooking radiometers' field of view, an adjustable boom was installed in 2015 Sep. The boom is kept about 1 m above the snow surface to eliminate non-snow objects from the radiometers' field of view. Thus, for water years 2016 and 2017, the adjustable downlooking boom measurements were used, while for all other years, only the fixed boom downlooking measurements could be used. We could not find a good relationship between the fixed boom and downlooking boom radiation values to correct the prior years. Instead, we used a bias correction based on the maximum observed annual albedo, explained below.

Because of the issue of not being able to discriminate diffuse radiation from the sky versus diffusion radiation from the snow and problems with measurements at high solar zenith angles, albedo measurements were only used 1x/day during clear sky conditions, $D_\downarrow/(B_\downarrow + D_\downarrow) \leq 0.2$, around solar noon. In addition, to eliminate problems with shadows, only the maximum downlooking daily values were retrieved. Thus, the database was queried to find the maximum daily downlooking broadband radiation values during clear sky conditions and to return the associated time, uplooking broadband/nIR measurement, and downlooking nIR measurement. Queries were further restricted to times when at least 30 cm of snow was measured on the ground and direct solar radiation was > 400 W m$^{-2}$ to ensure that only sunny days with broad snow cover were selected. The uplooking direct and nIR measurements were then corrected to the snow surface using a correction factor $c$ (Bair et al., 2015; Painter et al., 2012):

$$c = \frac{\cos \theta}{\cos \theta_0} \tag{2}$$

with $\theta$ as the local solar illumination angle such that the albedo $\alpha$ is the ratio of the reflected solar radiation $D_\uparrow$ and the terrain-corrected direct and diffuse solar radiation, with the diffuse being uncorrected assuming a negligible terrain effect on it:

$$\alpha = \frac{D_\uparrow}{cB_\downarrow + D_\downarrow}. \tag{3}$$

For the uplooking near infrared measurements, the diffuse fraction was not known, so we assumed that all of the radiation was direct based on atmospheric scattering being small at these wavelengths.

To compute $\theta$, the slope and aspect of the snow surface must be measured. To do this we used a Reigl Z390i laser scanner that operates automatically on a schedule at CUES that has varied over the study period from every 15 min to every hr. A point cloud from the scan nearest the albedo acquisition was selected. Then, a 2 m$^2$ bounding box within both the fixed and the adjustable downlooking radiometers' fields of view was used to filter this point cloud. This filtered point cloud was then fit with a plane to



determine the local slope and aspect. There were periods when the laser scanner was not working properly or times prior to its installation in 2011 Feb. For these times, the modal aspect (north) and slope (4º) were used for the terrain correction. Because the albedos were measured near solar noon and the slope of the terrain is low, $c$ is significantly less than one only during times when $\theta_0$ is high; those are times around solar noon during the accumulation season. From mid-April through melt out, $c$ is close to one,

making the terrain correction negligible.

From the corrected albedos, for each season, the maximum value for that season was compared with a theoretical maximum of 0.89 for a broadband albedo and 0.74 for an nIR albedo (Dozier et al., 2009). Values were then adjusted up or down by the difference between the observed and theoretical maximum on an annual basis. The average annual correction was +5.1% for the broadband albedo and -3.9% for the nIR albedo. These values suggest that trees, which are darker in the visible spectrum but

brighter than coarse grained dirty snow in the nIR, were often in the downlooking radiometers' field of view, especially later in the season. A minimum theoretical albedo was not used for correction as this will vary from season to season (e.g. Painter et al., 2012) depending on the concentration of impurities on the surface of the snowpack.

The assumption behind our maximum albedo correction is that the annual calibration or swapping of some of the instruments that occurs at CUES each fall could explain the bias and that this bias is scalar in nature. The latter assumption is unlikely to be true,

but we decided it was the best correction given the documented radiometer biases (Wilcox and Myers, 2008). We note that WY 2017 required a negligible correction since the adjustable downlooking boom was used. Curiously, albedos from WY2016 required a negative correction even though this was the first year that the adjustable downlooking boom was installed. Our explanation is that between WY 2016 and WY 2017 the downlooking boom design changed such that aluminum from the downlooking boom was visible to the radiometers on the downlooking boom in 2016 but not in 2017. Reflected light from the aluminum boom caused

the downlooking radiometers to have high biased measurements. Also, we note that the spectral range of the SPN-1 and the PSP are different, 0.400 to 2.700 vs. 0.285 to 2.800 μm (Table 2), however because of different documented biases (Wilcox and Myers, 2008), especially at higher solar zenith angles, the SPN-1 shows 2.5% more broadband radiation on average than the PSP. This sort of unanticipated bias further supports our approach of using a theoretical maximum albedo to bias correct our measurements.

For the times when the uplooking radiometers at CUES were not working, albedos were estimated using a multivariate regression

based on time since last snowfall (of at least 2.54 cm/1 inch) and $\theta$. This approach showed similar accuracy to what was expected from a simple statistical model based only on snowfall and solar geometry, $r^2=0.62$, RMSE=7.0%. Other variables such as new snow density and total snow depth were added to the regression but did not improve its accuracy. We note that that this RSME value is only slightly larger than the average annual bias correction of 5.1%, illustrating the uncertainties associated with in situ snow albedo measurement.

From these broadband and near-infrared albedos, the grain size and impurity content of the snowpack were estimated using a two-stream radiative transfer model where grain size and impurity content are solved simultaneously using nonlinear optimization (Meador and Weaver, 1980; Moré, 1977). The grain sizes and impurity content were then interpolated from the daily to hourly times across the study period. This grain size interpolation likely overestimates grain sizes on days with new snowfall that were cloudy, as no albedo measurements were taken on these days. One approach would be to use a constant value for new snow albedo,

which is done for age-based models (e.g. Dickinson et al., 1993; U.S. Army Corps of Engineers, 1956), however, on the days with new snowfall of greater than 2.54 cm/1 in that were sunny and therefore had albedo measurements around solar noon, there was substantial scatter in the new snow albedo, ranging from 0.60 to 0.89, illustrating the pitfalls of using a constant new snow albedo.



The interpolated grain sizes and impurity content were then fed back into the model varying $\theta_0$ with each hour as the solar zenith changed. The assumptions of this approach are that 1) the grain size changes relatively less than the albedo throughout the diurnal cycle; and 2) the daily albedo cycle is more accurately modeled than measured, which is based on our experience at CUES. Assumption 1) is the weaker assumption as grain size decay or growth can be rapid in the first day or two after snowfall (Flanner

and Zender, 2006) but for snow that is older than a day or two, which is most melting snow, the assumption is reasonable.

## 4 Results and discussion

We've selected five different measurement areas for comparison: snow depth and air temperatures at Sesame and CUES; albedo cycle at CUES; wind climatology at CUES; and uplooking longwave radiation at CUES.

### 4.1 Snow depth

The 2011 to 2017 water years show a tremendous diversity in snow accumulation. At Sesame, where precipitation records go back to the 1983 water year, 2017 was the wettest year on record with 2548 mm of precipitation, while 2015, with 548 mm of precipitation, was the driest. We stress that the measurements from Sesame do not cover the entire water year and do not always cover consistent time periods from year-to-year, depending on when the ski resort opened and closed. An examination of the nearby Mammoth Pass (CDEC code MHP) snow course, with records back to 1928, shows two years with greater maximum SWE on the

ground: 2197 mm of SWE on 1969-3-27 and 2159 mm on 1983-4-25. Water year 2017 was third with 2083 mm of SWE on the ground on 2017-3-31. There are no reliable precipitation gauges with record lengths > 40 years in the area around Mammoth Mountain. The closest is Huntington Lake, lower at 2134 m elevation, where 2017 ranks 5[th] among water years going back to 1912.

In terms of maximum snow depth, 2015 was the lowest on record with 75 cm at Sesame and 112 cm at CUES. Despite having the most precipitation, water year 2017, with peak base depths of 526 cm at Sesame and 543 cm at CUES did not have the deepest

snow depths recorded at either site. At Sesame, 2006, with a 610 cm peak base depth and 1995 with 561 cm peak base depth, both had more snow on the ground than 2017. At CUES, reliable base depths only go back to 2001, but snow depth was over 600 cm in 2006 when the downlooking boom was buried. Subsequently, it was raised up to the top of the railing from the plaform floor. Overall, the snow depth at both sites agree well (Figure 4) with CUES having a later melt out date, explained by its higher elevation and slightly north facing terrain, while Sesame had slightly greater snow depths in the wet years, i.e. 2011 and 2017, possibly

because the snow under the depth sensor at CUES was removed by wind transport during the big storms.

### 4.2 Air temperature

The Sesame site is slightly warmer than CUES with an average annual temperature of 4.88 ºC vs. 4.50 ºC, although the November through May temperatures, which corresponds to the average period when snow is on the ground, are equivalent to within the instrument uncertainty with both sites at -0.11 ºC. Comparing midwinter temperatures at both sites (Figure 5), we see that above

freezing temperatures are common and that the diurnal range is considerably narrower at CUES, which follows given its mid-mountain and exposed location in comparison to Sesame's location near a valley, with Seame subject to longwave heating from the trees and cold air pooling at night.





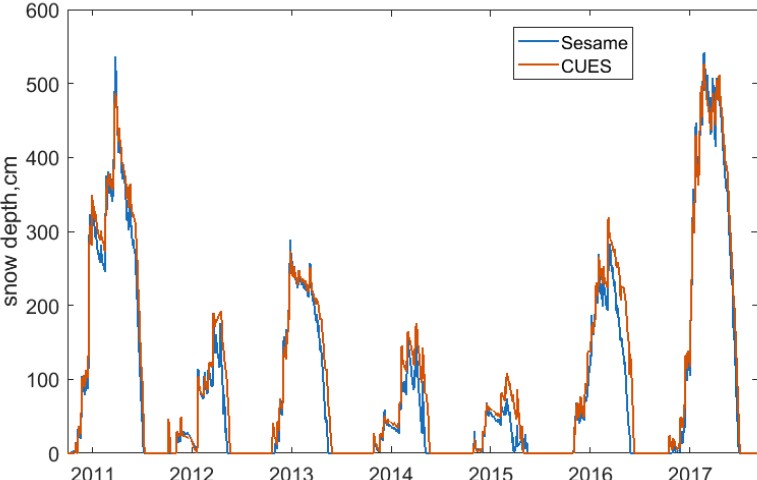

**Figure 4.** Snow depth at Sesame and at CUES during the study period. Note the similar depths with slightly later melt out at CUES, especially in the drier years.

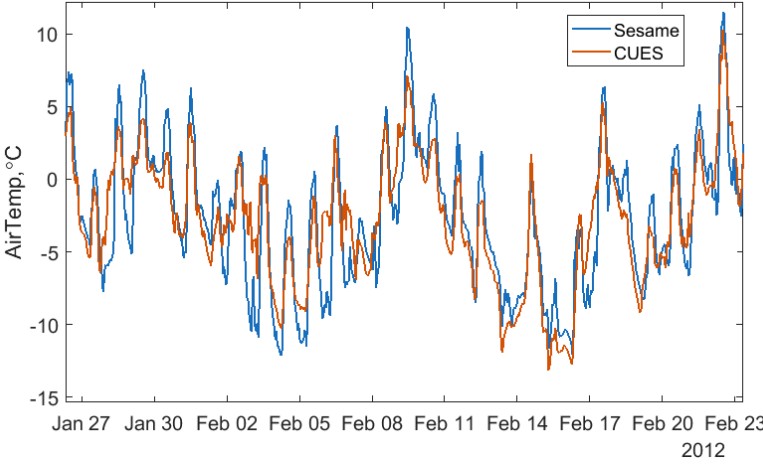

**Figure 5.** Air temperature at Sesame and at CUES during a selected midwinter period. Note the greater diurnal range for Sesame.

### 4.3 Albedo

The broadband snow albedo at CUES is usually above 0.80 for much of the accumulation season with values reaching above 0.90 for the highest solar zenith angles. When the albedo stops being refreshed by new snow and the old snow is covered with pumice, the albedo drops dramatically. In every season, minimum albedo values were < 0.60. A large diurnal variation in albedo of > 20% is evident for days late in the melt season due to the range of solar zenith angles (Figure 6). Although there is little energy reaching the snowpack in the early morning and late afternoon, when the solar zenith angles and albedo are highest, the illumination angle effect on albedo is significant and should be included in all snowmelt models nonetheless.



### 4.4 Wind

A wind rose for CUES (Figure 7) shows wind speeds in the range of previously published measurements from an anemometer mounted on top of Main Lodge (Bair, 2013). As with all mountain areas, there is substantial variability in the wind speed. For example, the average ridge top wind speeds are at least 50% greater than these (Bair, 2011) and the highest reliably recorded wind

5      gust from the top of Mammoth Mountain was measured at 82.3 m sec$^{-1}$.

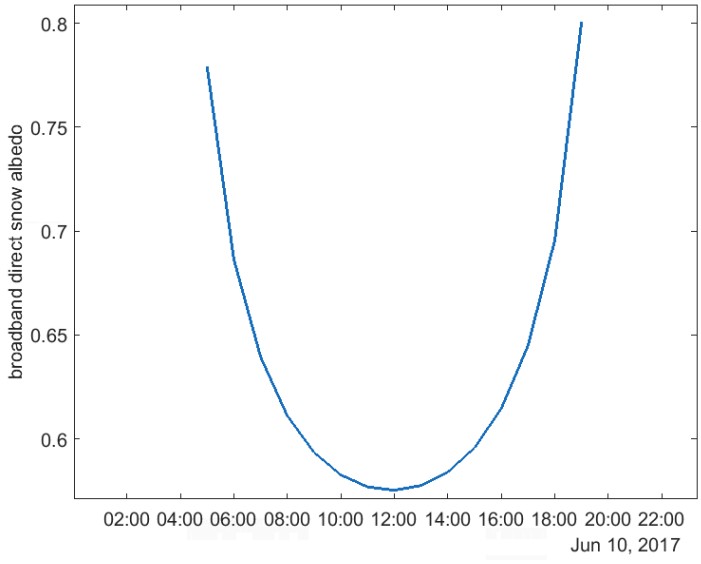

**Figure 6. Diurnal variation in albedo for a day in 2017 Jun. Note the large range, although the solar energy reaching the snowpack at the times with the highest albedo is low.**

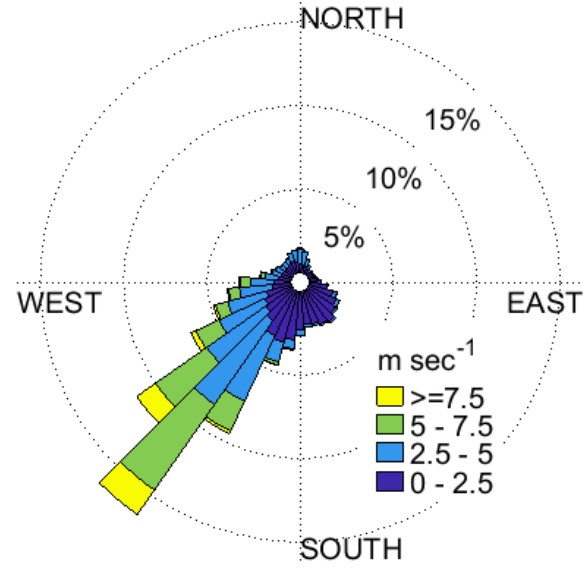

10      **Figure 7. Wind rose for CUES, a windy site. Note the predominant southwest wind direction.**



### 4.5 Uplooking longwave radiation

To illustrate the importance of measuring incoming longwave radiation rather than modeling it using the more commonly available temperature and relative humidity measurements, we've plotted modeled values against measured values for clear sky conditions during the day (Figure 8), since the model (Marks and Dozier, 1979) is optimal for clear sky conditions, with clear conditions

defined the same as for direct albedo measurement: $D_\downarrow/(B_\downarrow + D_\downarrow) \leq 0.2$. There is a strong negative bias of -40 W m$^{-2}$ or -17% of the mean measured value. The RMSE of 43 W m$^{-2}$ is within the ranges reported in Marks and Dozier (1979) and they also report similar negative bias.

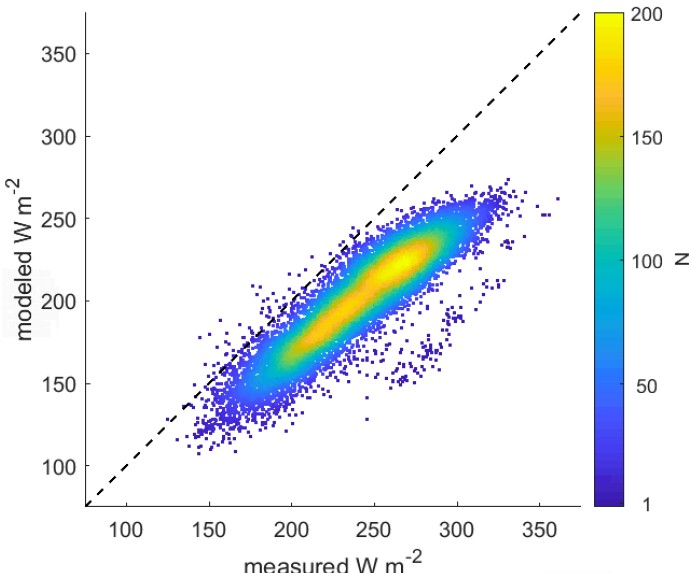

**Figure 8. Density plot of modeled vs. measured uplooking longwave radiation for clear sky measurements. Number of measurements $N$**

**is represented by the color scale. Note the clear negative model bias.**

### 5 Conclusion

In order to provide hourly energy balance measurements that can be used in a variety of snow models, we have created a carefully filtered dataset using instruments at two sites on Mammoth Mountain, CA. These years comprise one of the wettest and driest years since 1983. Unique measurements include hand weighed daily snow measurements from the Sesame Snow Study Plot and

terrain corrected broadband snow albedo. This dataset only comprises a fraction of the measurements available on Mammoth Mountain. We encourage interested researchers to explore the raw measurements available on the CUES website at www.snow.ucsb.edu if this dataset does not meet their modeling needs.

### 6 Data availability

These data are available at www.snow.ucsb with doi: 10.21424/R4159Q. They consist of three large comma separated tables,

uncompressed in ASCII format with one line headers. The tables are: the daily Sesame Snow Study Plot manual precipitation and weather with notes; the hourly Sesame Snow Study Plot Air Temperature, Relative Humidity, and Snow Depth; and the CUES hourly radiation, snow albedo, windspeed, air temperature, relative humidity, air pressure, and snow depth.



## 7 Supplement link

(to be provided by Copernicus)

## 8 Author contribution

Dr. Bair performed most of the data production, analysis, and authoring of the manuscript. Prof. Dozier wrote the radiative transfer
incoming solar filtering code. He and Dr. Davis, who both edited the manuscript, have kept CUES funded and running in its current
location for almost 30 years.

## 9 Competing interests

The authors declare that they have no conflict of interest.

## 10 Acknowledgements

This work was supported by NASA awards NNX12AJ87G and NNX15AT01G and U.S. Army Cold Regions Research and
Engineering Laboratory award W913E5-16-C-0013.

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



## 12    Tables

**Table 1. List of measurements and instruments used in this dataset at the Sesame Snow Study Plot**

Instrument specifications are from the manufacturer unless noted and cited

| Measurement | Instrument | Key instrument specifications |
|---|---|---|
| Total snow depth | Ultrasonic Judd Depth Sensor | Target to bare ground distance: 619 cm<br>Relative accuracy ± 0.4%, thus 2.5 cm accuracy at worst<br>22º beam width, footprint size 236 cm at most |
| 24-hr new snow, automated | Ultrasonic Judd Depth Sensor | see above |
| 24-hr new snow/SWE, manual | Snowmetrics 12" tube and spring scale | Observer and snowpack type dependent, but widely considered to be the most accurate method of measuring new snow and SWE |
| Air temperature/relative humidity | Campbell Scientific HMP45C | ± <0.4 ºC accuracy for air temperature and ± 3% for relative humidity |

5  **Table 2. List of measurements and instruments used in this dataset at CUES**

| Measurement | Instrument | Key instrument specifications |
|---|---|---|
| Total snow depth | Campbell Scientific SR50 | Target to bare ground distance: 617 cm<br>Accuracy ± 0.4%, yielding 2.5 cm accuracy<br>22º beam width, footprint size 235 cm at most |
| Air temperature/relative humidity | Campbell HMP45C | See above |
| Air pressure | Campbell Scientific CS100 | Accuracy: ± 1e-3 mb |
| Upward looking direct broadband solar radiation | Delta-T Sunshine Pyranometer SPN1 | Spectral response: 0.400 to 2.700 μm<br>Manufacturer accuracy: not given<br>Wilcox and Myers (2008) Feb to May accuracy: 3.0 to 8.1% bias |
| Upward looking diffuse broadband solar radiation | Delta-T Sunshine Pyranometer SPN1 | Spectral response: 0.400 to 2.700 μm<br>Manufacturer accuracy: ± 5.0%<br>Wilcox and Myers (2008) Feb to May accuracy: -13.8 to -4.3% bias |
| Upward looking near infrared solar radiation | Ventilated Eppley Precision Spectral Pyranometer with Schott glass RG8 hemispherical filter | Spectral response: 0.700 to 2.800 μm<br>Accuracy: ± 2.0 % |
| Downward looking radiation | Eppley Precision Spectral Pyranometer with Schott glass WG7 clear dome | Spectral response: 0.285 to 2.800 μm<br>Accuracy: unknown for diffuse radiation from snow |
| Downward looking near infrared radiation | Eppley Precision Spectral Pyranometer with Schott glass RG8 hemispherical filter | Spectral response: 0.700 to 2.800 μm<br>Accuracy: unknown for diffuse radiation from snow |
| Upward looking longwave radiation | Eppley Precision Infrared Radiometer | Spectral response: 4.00 to 50.00 μm<br>Accuracy: ± 2.5% |
| Wind speed and direction | RM Young 5103 Wind Monitor | Range: 1.1 to 100 m/s<br>Accuracy: speed ± 1 %; direction ± 3º |
|  | Lufft WS600 UMB ultrasonic anemometer | Range: 0 to 75 m/s<br>Accuracy: speed ± 1 m/sec; direction ± 3º |