# Peer review of "Hourly mass and snow energy balance measurements from Mammoth Mountain, CA USA, 2011-2017"

_Earth System Science Data, 2017_

## Referee Comment (RC1) · Anonymous Referee #1 · 15 Nov 2017

Bair et al. present a six year dataset of snow and energy balance measurements at Mammoth Mountain, California. The data include: (1) daily precipitation and hand-weighed SWE at the Sesame Street Snow Study Plot, (2) hourly temperature, relative humidity, and snow depth at Sesame, and (3) hourly uplooking shortwave, longwave, albedo, air temperature, wind speed/direction, relative humidity, air pressure, and snow depth data at the nearby CUES site. The authors describe the data sources, instruments, and processing routines and discuss a subset of variables over the presented record (water years 2011-2017), which include extreme wet and dry years.

Given the notable lack of energy balance measurements in the Sierra Nevada, I think this dataset fills a clear gap and would be useful to the community. I recommend publishing it in the journal after attention to the comments below.

[Figure]

COMMENTS

- My most major comment is that I think the dataset may have limited usefulness for evaluating snow models. The authors argue that the dataset is useful for running models (e.g., abstract and introduction) and it is true that they are providing all required data to do so (e.g., temperature, humidity, wind, precipitation, radiation). However, the main problem with the dataset is that it has the bare minimum in evaluation data. Depending on whether the albedo is used as a model input or model evaluation dataset, there are only two or three datasets to check the snow model (e.g., hourly snow depth, event-based SWE from hand measurements), and that will only provide limited insights into model behavior (in my opinion). The hand measurements of SWE and snow density are really only useful at the time of the storm event, and provide no information about what is happening to the existing snowpack in time. Hence, the presented data really provides no direct way of checking the model representation of the mass balance and energy states through time, and other data would be needed (e.g., snow pillow SWE, snow surface temperature, snow pit profile data, etc.). Unless the authors are willing to include the snow pillow data (albeit incomplete) and any other relevant evaluation data, I am not really sure how this problem can be adequately remedied. At a minimum, the authors should at least detail ideas on how the dataset could be used not only to run models but also to evaluate them, given minimal evaluation data. A direct demonstration with a snow model and the dataset would be instructive and would match other snow data papers.

- Introduction: It would be useful to identify other similar snow datasets available (and cite the data papers), for example at Reynold's Mountain, Senator Beck Basin, and others. Highlighting the unique attributes of Mammoth relative to these areas would be helpful to the community.

- I think the paper would be more useful if more specific guidance/recommendations were provided to scale the daily precipitation data to hourly. This is not trivial, given that mixed precipitation and rain are possible and hence assuming uniform precipitation

over all 24 hours is not necessarily a robust approach.

- While the snow pillow measurements do not span the entire period, I think they still hold enough value that they should be included in the dataset, without having to request from the authors. For long-term purposes, it would be more ideal if researchers ten years from now do not have to track down the authors to obtain these snow pillow data.

TECHNICAL CORRECTIONS

- P2.L4-6: While interesting, this would be more relevant if you actually detected any such events in the snow albedo dataset. Please comment.

- P.2, L.7-9: Given the winter recreation, please comment on what measures (if any) were in place to prevent humans from impacting measurements (e.g., skiing underneath the snow depth sensor).

- P.2, L.20: Awkward wording here because the phrase "to accurately weigh" splits the phrase "snow falls". Please rephrase.

- P.3, L.1: "on as" reads oddly to me. Delete one word?

- P.3, L.7: Recommend starting a new sentence at the semicolon: "one minute readings. The measurements from these gauges...".

- P.9, L18-25: It is not clear what "peak base depth" or even "base depth" means. Please clarify.

- P.12, L1-4: This is a long sentence that really would be better framed as two sentences.

TABLE AND FIGURE COMMENTS

- Figure 10: Please confirm these are hourly values and the period of record included in this figure.

DATASET COMMENTS

[Figure]

- In the daily precipitation table, please include measurement units with each variable name in the header. Also, it may help to have a metadata file describing what each of the columns means, as there are some that I think are not necessary self-evident. It would be useful to have some guidance on how to use the precipitation data, as only days with precipitation appear to be recorded in the table. Is it safe to assume these are all 24 hour measurements and days not in this table have no precipitation?

- There is a value of 90% snow density on October 19, 2015 which is physically unlikely, as it is close to the density of pure ice. Please check.

- At both sites, there are RH values exceeding 1.0. Please provide additional quality control.

- At Sesame, there are wildly varying RH values in July 2017 and early August 2017. Please provide additional quality control.

---

## Referee Comment (RC2) · Anonymous Referee #2 · 21 Nov 2017

Review of 'Hourly mass and snow energy balance measurements from Mammoth Mountain, CA USA, 2011-2017' by E. Bair, R. Davis, and J. Dozier

The authors present a dataset of snowfall mass and energy balance compiled from two neighboring stations located in a snow-dominated mountainous environment in the eastern Sierra Nevada, California. The study site is described as one of only five energy balance monitoring stations in the Western U.S. The authors describe the dataset as useful to run a variety of snow models.

The dataset includes: 1) hourly air temperature, relative humidity, wind speed and direction, and air pressure, 2) hourly incoming shortwave and longwave radiation including shortwave direct and diffuse components, 3) hourly snow depth, 4) daily surface albedo, and 5) daily wintertime snowfall (hand measured snow water equivalent).

[Figure]

The quality of the seven-year (2011-2017) dataset is high. The paper is fairly well written and the methods and data are well-described. In my opinion, the strength of the dataset is in the availability of hourly shortwave radiation (the availability of both direct and diffuse components is rare), which provides substantial information on cloud cover, longwave radiation (required by energy balance snowmelt models), and albedo (useful to either force a snowmelt model or verify empirical algorithms within such models). These observations could benefit an array of Earth system sciences, including snow hydrology, remote sensing and land-atmosphere interactions. For that reason, I support the ultimate publication of the paper and dataset.

I have a few concerns that prevent me from recommending publication in the present form. The product lacks hourly precipitation necessary to run most snow energy balance models, and lacks snow water equivalent data necessary to validate a snow model. The title does not appropriately describe the dataset. Finally, the paper would benefit from 1) an expanded description of how these variables are used in Earth Sciences, 2) evidence of data quality (figures), and enhanced examples of its application. Such additions would greatly improve the paper and extend its utility across a range of Earth sciences. Please see my associated comments, below.

(i) Hourly precipitation data have become a standard requirement of snow energy balance models. The title 'hourly mass . . . balance' is misleading – only daily snowfall is provided and accurate snow mass balance typically requires all-phase precipitation including rainfall. Could regional (hourly) precipitation measurements (e.g., SNOTEL) be used to inform a temporal interpolation of daily hand-measured SWE to an hourly product? Providing an hourly precipitation product may support more diverse application and user interest.

Addressing the second point (mass balance) may be more difficult. Because hand-weighed SWE measurements are rare, and in the absence of local total precipitation measurements, the authors must better and more carefully explain what information these data contain and what information they may lack. What are the potential pitfalls

of using such measurements to constrain the snow mass balance in general and at this location (blowing snow, melting, rainfall)? Further, how does a plot of cumulative hand-measured snowfall compare to a time-series of seasonal SWE measured on the ground? The authors do not offer enough data to promote an understanding of hand-measured snowfall.

(ii) A more detailed description of common (potential) uses of this dataset would be helpful. It may be worth mentioning the utility of the dataset for validating distributed products such as NLDAS-2 or remote sensing products.

(ii) Even if the snow pillow SWE data are limited, they could still be a substantial resource for users looking to verify a snowmelt model. I strongly suggest that these data be included. The SWE is more directly relevant to the mass and energy balance theme strummed in the title than snow depth.

(iii) "A variety of snow models" on Line 15 is vague . . . I expected more discussion and examples in the paper. Along the lines of a paper by Landry et al. (2014) that highlight similar measurements from a site in Colorado, it would be helpful to include an example of snow model results forced and verified by the data. This would also serve as evidence of data quality and utility.

Landry, C. C., Buck, K. A., Raleigh, M. S., & Clark, M. P. (2014). Mountain system monitoring at Senator Beck Basin, San Juan Mountains, Colorado: A new integrative data source to develop and evaluate models of snow and hydrologic processes. Water Resources Research, 50(2), 1773-1788.

(iv) Please provide some discussion about the possibility of snow on the radiometer sensors, how these times might be flagged, and some words of caution.

---

## Referee Comment (RC3) · Anonymous Referee #3 · 23 Nov 2017

General comments

The paper summarizes data, data quality, and data cleaning and interpolation methods from sites operated by the University of California Santa Barbara on Mammoth Mountain, CA. Data from 2011 through 2017 is presented including a continuation of snow energy balance component measurements at the long-running "CUES" site. Coupled with newer intensive measurements of snow depth and density, in particular daily new snow depth and SWE hand measurements, make this a unique and valuable dataset. Figure 8 and associated discussion are particularly valuable contributions.

The manuscript contains much useful information, particularly in the methods section,

and it would benefit from a thorough editing to trim unnecessarily wordy sections and clarify meaning. Examples are mentioned in the technical correction section. Further, I found the organization of the Datasets section confusing. The first section is on energy balance components, while the second section focused on data filtering and processing. It would be better if the organization were by measurement type (energy balance components, and snow measurements) with methods for data processing and cleaning included under these general headings. Specific headings for each measurement type would also help guide the reader (wind speed / direction, temperature, albedo, etc).

Specific comments

1) Please justify using climatological averages to gap fill temperature, RH, and air pressure data. It seems that this would create steps in the data and impose periods of average conditions during periods when the climate was likely not average (and hence went off-line). Why not use linear regression with other sites in the area for gap fill periods greater than 12 hours?

2) The CUES site is on the north aspect of Mammoth Mountain. Does this topography impact the measurement of direct and diffuse radiation? If so, what is the timing and magnitude of the impact relative to a site without substantial topographic shading?

3) How was the WS600 data used to fill gaps in RM Young 5105 time series (regression, replacement, other)?

Technical corrections/comments:

Minor comment: Figure 3 would be more effective if it were side-by-side with a snow-off photo and/or with arrows pointing to the instrumentation referenced in the caption.

Section 1 Introduction. I suggest replacing "tedious and nontrivial adjustments that are only possible by those intimately familiar . . ." be replaced with something like "substantial and nontrivial adjustments that require detailed information on measurement location characteristics". This is an important subject for all folks in the business of

collecting environmental data.

Section 2.1 and 2.2. Each section would benefit from an introductory paragraph describing the measurements at the site included in the dataset. It is difficult to sort through what is and is not in the dataset. Separating them would help.

Section 3.2.2 Albedo calculation. This is an important section that would benefit from some reorganization, specifically an introduction paragraph that outlines the overall process (e.g. "To calculate albedo, we did 1,2,3,4,5 etc").

Section 4.1 Moving the description of the data and mention of Figure 4 to the first couple of sentences would greatly clarify this section. Subsequent description of other years outside of the dataset then are placed in context.

---

## Author Comment (AC1) · 10 Jan 2018

We thank Referee #1 for his or her timely and constructive comments. Referee #1 and #2 had some similar comments which we will address first.

A shared critique is that the dataset should include hourly SWE. To this end, we will include the hourly snow pillow measurements from CUES (water years 2012 to 2017) and Sesame (water years 2013 to ~ Feb 2017, when the pressure transducer failed). Initially, we focused on providing a complete hourly dataset over the study period, however the referee comments have convinced us to include these incomplete hourly SWE measurements.

To provide a complete record of hourly SWE over the study period, we have also decided to include snow pillow, snow depth, and tipping bucket precipitation from the nearby Mammoth Pass (CDEC code MHP) station. MHP is not located within the ski area, as CUES and Sesame are, but is just outside the boundary and receives very similar precipitation amounts to what is recorded at Sesame. We will present a correlation analysis of the two sites to justify this statement for times when good hourly precipitation was available from Sesame. MHP is at an elevation between CUES and Sesame (2835 m), but is in a forested area similar to the Sesame site, thereby eliminating some of the wind exposure problems at CUES for measuring precipitation. Because it is in a forested area, it likely has (though there are no radiometric measurements to confirm this), a similar radiation budget to Sesame, but not CUES.

As already noted in the manuscript, the precipitation measurements at CUES and Sesame have substantial shortcomings. In fact, upon examination of the Sesame data, we found long periods when the tipping bucket heater was malfunctioning during the 2011 and 2012 water years, eventually requiring replacement from the manufacturer. Likewise, as noted already in the manuscript, the pressure transducer failed at Sesame in 2017, likely due to the exceptional weight of the snowpack.

The MHP measurements are not without their own problems, which we plan to discuss extensively. For example, the tipping bucket data suffer from a number of issues including gaps–likely when the orifice was clogged or the satellite modem could not transmit–causing precipitation to jump and show late timing when compared to the manually weighed precipitation at Sesame. We tried a number of approaches to address these problems, but many could not be fixed. These inherent deficiencies demonstrate the problems with using automated hourly precipitation gauge measurements at a snowy site and reinforce the value of the manual snow measurements from Sesame.

The other main critique of Referee #1, shared with Referee #2, is that a demonstration of a snow model, forced with these measurements is needed. This brings up the validation loop issue brought up by Referee #1 and #2. With the snow pillow measurements provided at CUES, we aim to close the forcing/validation loop using a demonstration with the widely used SNOWPACK model. For WY 2012-2017, we will model the snow mass balance at CUES, validated with the snow pillow measurements. We will force the model with the radiometric measurements at CUES and use 3 different precipitation forcings: a) hourly snow depth measurements over the pillow at CUES using SNOWPACK's empirical new snow density estimate; b) hourly tipping bucket precipitation measurements from MHP; c) daily manual SWE measured at Sesame scaled to hourly measurements, with knowledge of timing based on the tipping bucket measurements and automated snow depth measurements. This approach will address the issues of not just assuming uniform precipitation over 24 hr and of precipitation phase, both brought up by Referee #1.

We provide in-line responses to the individual points below.

Anonymous Referee #1

Bair et al. present a six year dataset of snow and energy balance measurements at Mammoth Mountain, California. The data include: (1) daily precipitation and hand- weighed SWE at the Sesame Street Snow Study Plot, (2) hourly temperature, relative humidity, and snow depth at Sesame, and (3) hourly uplooking shortwave, longwave, albedo, air temperature, wind speed/direction, relative humidity, air pressure, and snow depth data at the nearby CUES site. The authors describe the data sources, instruments, and processing routines and discuss a subset of variables over the presented record (water years 2011-2017), which include extreme wet and dry years.

Given the notable lack of energy balance measurements in the Sierra Nevada, I think this dataset fills a clear gap and would be useful to the community. I recommend publishing it in the journal after attention to the comments below.

- My most major comment is that I think the dataset may have limited usefulness for evaluating snow models. The authors argue that the dataset is useful for running mod- els (e.g., abstract and introduction) and it is true that they are providing all required data to do so (e.g., temperature, humidity, wind, precipitation, radiation). However, the main problem with the dataset is that it has the bare minimum in evaluation data. Depending on whether the albedo is used as a model input or model evaluation dataset, there are only two or three datasets to check the snow model (e.g., hourly snow depth, event- based SWE from hand measurements), and that will only provide limited insights into model behavior (in my opinion). The hand measurements of SWE and snow density are really only useful at the time of the storm event, and provide no information about what is happening to the existing snowpack in time. Hence, the presented data really provides no direct way of checking the model representation of the mass balance and energy states through time, and other data would be needed (e.g., snow pillow SWE, snow surface temperature, snow pit profile data, etc.). Unless the authors are willing to include the snow pillow data (albeit incomplete) and any other relevant evaluation data, I am not really sure how this problem can be adequately remedied. At a minimum, the authors should at least detail ideas on how the dataset could be used not only to run models but also to evaluate them, given minimal evaluation data. A direct demonstration with a snow model and the dataset would be instructive and would match other snow data papers.

See above

- Introduction: It would be useful to identify other similar snow datasets available (and cite the data papers), for example at Reynold's Mountain, Senator Beck Basin, and others. Highlighting the unique attributes of Mammoth relative to these areas would be helpful to the community.

Ok, we will do this

- I think the paper would be more useful if more specific guidance/recommendations were provided to scale the daily precipitation data to hourly. This is not trivial, given that mixed precipitation and rain are possible and hence assuming uniform precipitation over all 24 hours is not necessarily a robust approach.

Ok, and see initial paragraph

- While the snow pillow measurements do not span the entire period, I think they still hold enough value that they should be included in the dataset, without having to request from the authors. For long-term purposes, it would be more ideal if researchers ten years from now do not have to track down the authors to obtain these snow pillow data.

Ok, and see initial paragraph

TECHNICAL CORRECTIONS
- P2.L4-6: While interesting, this would be more relevant if you actually detected any such events in the snow albedo dataset. Please comment.

Even with very large grain sizes and a nadir solar zenith angle, clean snow albedo does not drop below 0.6 (Dozier et al., 2009). We show albedos below 0.6 every year, meaning surface impurities are be present. Sterle et al. (2013) confirm that the surface impurities at CUES are dust and black carbon.

- P.2, L.7-9: Given the winter recreation, please comment on what measures (if any) were in place to prevent humans from impacting measurements (e.g., skiing underneath the snow depth sensor).

Ok, we will mention that both sides are roped off and signed

- P.2, L.20: Awkward wording here because the phrase "to accurately weigh" splits the phrase "snow falls". Please rephrase.

Ok, we will fix.

- P.3, L.1: "on as" reads oddly to me. Delete one word?  - P.3, L.7: Recommend starting a new sentence at the semicolon: "one minute readings. The measurements from these gauges. . .".  -

Ok, we will fix.

P.9, L18-25: It is not clear what "peak base depth" or even "base depth" means. Please clarify.

Ok, will change to "peak snow depth".

- P.12, L1-4: This is a long sentence that really would be better framed as two sen- tences.

Ok, we will fix.

TABLE AND FIGURE COMMENTS
- Figure 10: Please confirm these are hourly values and the period of record included in this figure.

There is no figure 10. Figure 7 maybe? We will add that these are hourly values for the period of record.

DATASET COMMENTS

- In the daily precipitation table, please include measurement units with each variable name in the header.

Ok, we will fix, good point. We should note that we've kept only these hand weighed measurements in Imperial units, since they were taken this way, and the notes often refer to the measurements in these units.

Also, it may help to have a metadata file describing what each of the columns means, as there are some that I think are not necessary self-evident.

Ok, we will create a file to describe the column headers.

It would be useful to have some guidance on how to use the precipitation data, as only days with precipitation appear to be recorded in the table. Is it safe to assume these are all 24 hour measurements and days not in this table have no precipitation?

This is already addressed in the text on p 2, l 20

"We provide all the manual Sesame Snow Study Plot measurements (Table 1) for days with precipitation, based on the morning daily weather observations, posted on as the "Storm Summaries" on http://patrol.mammothmountain.com."

- There is a value of 90% snow density on October 19, 2015 which is physically unlikely, as it is close to the density of pure ice. Please check.

The reported "density" is simply the water equivalent (WE) / New Snow (HN), in this case (0.45" WE/0.5" HN). In the previously suggested metadata, we will clarify this.

- At both sites, there are RH values exceeding 1.0. Please provide additional quality control.

RH can and should exceed 1.0 regularly at this site.

- At Sesame, there are wildly varying RH values in July 2017 and early August 2017. Please provide additional quality control.

Ok, we will fix those values.

Dozier, J., Green, R.O., Nolin, A.W. and Painter, T.H., 2009. Interpretation of snow properties from imaging spectrometry. Remote Sensing of Environment, 113: S25-S37.
Sterle, K.M., McConnell, J.R., Dozier, J., Edwards, R. and Flanner, M.G., 2013. Retention and radiative forcing of black carbon in eastern Sierra Nevada snow. The Cryosphere, 7: 365-374.

---

## Author Comment (AC2) · 10 Jan 2018

We thank Referee #2 for his or her thoughtful critique. Two of the main criticisms were also brought up by Referee #1, therefore we will refer Referee #2 to our response to Referee #1 in several places. We will respond to each of the comments below in-line.

The authors present a dataset of snowfall mass and energy balance compiled from two neighboring stations located in a snow-dominated mountainous environment in the eastern Sierra Nevada, California. The study site is described as one of only five energy balance monitoring stations in the Western U.S. The authors describe the dataset as useful to run a variety of snow models.
The dataset includes: 1) hourly air temperature, relative humidity, wind speed and direction, and air pressure, 2) hourly incoming shortwave and longwave radiation including shortwave direct and diffuse components, 3) hourly snow depth, 4) daily surface albedo, and 5) daily wintertime snowfall (hand measured snow water equivalent).

The quality of the seven-year (2011-2017) dataset is high. The paper is fairly well written and the methods and data are well-described. In my opinion, the strength of the dataset is in the availability of hourly shortwave radiation (the availability of both direct and diffuse components is rare), which provides substantial information on cloud cover, longwave radiation (required by energy balance snowmelt models), and albedo (useful to either force a snowmelt model or verify empirical algorithms within such models). These observations could benefit an array of Earth system sciences, including snow hydrology, remote sensing and land-atmosphere interactions. For that reason, I support the ultimate publication of the paper and dataset.

I have a few concerns that prevent me from recommending publication in the present form. The product lacks hourly precipitation necessary to run most snow energy balance models, and lacks snow water equivalent data necessary to validate a snow model.

This is an excellent point, and one was also brought up by Referee #1. We will include the hourly SWE measurements from CUES as well as hourly precipitation and SWE from MHP. See our response to Referee #1. Concerning the hourly precipitation measurements, we note that there are many insurmountable problems with using gauges to record precipitation in snowy areas. Inclusion of the hourly gauge precipitation from MHP will demonstrate these problems further, although we agree that they should still be included.

The title does not appropriately describe the dataset.

With the inclusion of the hourly SWE and tipping bucket measurements, the title will be appropriate.

Finally, the paper would benefit from 1) an expanded description of how these variables are used in Earth Sciences,

Ok, we will further discuss the use of these measurements in Earth Sciences in the Introduction.

2) evidence of data quality (figures), and enhanced examples of its application.

We will provide figures and examples of its application by using a snow model (SNOWPACK) forced with our radiative measurements from CUES + different precipitation forcings and verified with snow pillow measurements at CUES. See our responses to Referee #1.

Such additions would greatly improve the paper and extend its utility across a range of Earth sciences. Please see my associated comments, below.

Hourly precipitation data have become a standard requirement of snow energy balance models. The title 'hourly mass . . . balance' is misleading – only daily snowfall is provided and accurate snow mass balance typically requires all-phase precipitation including rainfall. Could regional (hourly) precipitation measurements (e.g., SNOTEL) be used to inform a temporal interpolation of daily hand-measured SWE to an hourly product? Providing an hourly precipitation product may support more diverse application and user interest.

Excellent points. Please see our response to the main critique and to Referee #1.

Addressing the second point (mass balance) may be more difficult. Because hand-weighed SWE measurements are rare, and in the absence of local total precipitation measurements, the authors must better and more carefully explain what information these data contain and what information they may lack. What are the potential pitfalls of using such measurements to constrain the snow mass balance in general and at this location (blowing snow, melting, rainfall)?

Further, how does a plot of cumulative hand-measured snowfall compare to a time-series of seasonal SWE measured on the ground? The authors do not offer enough data to promote an understanding of hand-measured snowfall.

Good points. We suggest these concerns will be addressed with the inclusion of the SWE and tipping bucket measurements, which will also allow the daily manual measurements to be better scaled to hourly measurements. We want to emphasize that hand weighed snowfall is the gold standard for measuring solid precipitation. Gauges, especially in the presence of wind, perform poorly (e.g. Rasmussen et al., 2012). We will more thoroughly discuss the advantages and disadvantages of gauges, pillows, and manual snow measurements.

(ii) A more detailed description of common (potential) uses of this dataset would be helpful. It may be worth mentioning the utility of the dataset for validating distributed products such as NLDAS-2 or remote sensing products.

We will discuss potential uses further. Validation of remote sensing products such as NLDAS-2 at 1/8º with point data present serious point to area extrapolation problems. Point snow model validation and possibly validation of higher resolution remotely-sensed products such as snow albedo from LandSat (Painter et al., 2009; Rittger et al., 2013) are more appropriate.

(ii) Even if the snow pillow SWE data are limited, they could still be a substantial resource for users looking to verify a snowmelt model. I strongly suggest that these data be included. The SWE is more directly relevant to the mass and energy balance theme strummed in the title than snow depth.

Agree and will be done, as previously noted several times.

(iii) "A variety of snow models" on Line 15 is vague ... I expected more discussion and examples in the paper. Along the lines of a paper by Landry et al. (2014) that highlight similar measurements from a site in Colorado, it would be helpful to include an example of snow model results forced and verified by the data. This would also serve as evidence of data quality and utility.

Agree and will be done, as previously noted.

Landry, C. C., Buck, K. A., Raleigh, M. S., & Clark, M. P. (2014). Mountain system monitoring at Senator Beck Basin, San Juan Mountains, Colorado: A new integrative data source to develop and evaluate models of snow and hydrologic processes. Water Resources Research, 50(2), 1773-1788.

(iv) Please provide some discussion about the possibility of snow on the radiometer sensors, how these times might be flagged, and some words of caution.

At CUES, the SPN-1 radiometer is heated, thus we have not observed it being snow covered. For the missing uplooking radiation filled in from DAN, snow cover over the radiometer is possible. This condition can be diagnosed when the downlooking DAN radiation is greater than the uplooking radiation. Theoretically, the "net solar" radiation measurement from DAN would be negative during these times, however this measurement is clearly erroneous, with large negative values most days of the year during the study period, so it cannot be used.

Thus, we will mention that the DAN radiometer could have been snow covered, but that this is a minor issue. To put the problem in context, DAN measurements during snowfall (which is when the radiometer would most likely be covered) only comprise 0.26% of the hourly uplooking solar measurements.

Painter, T.H., Rittger, K., McKenzie, C., Slaughter, P., Davis, R.E. and Dozier, J., 2009. Retrieval of subpixel snow-covered area, grain size, and albedo from MODIS. Remote Sensing of Environment, 113: 868-879.
Rasmussen, R., Baker, B., Kochendorfer, J., Meyers, T., Landolt, S., Fischer, A.P., Black, J., Thériault, J.M., Kucera, P., Gochis, D., Smith, C., Nitu, R., Hall, M., Ikeda, K. and Gutmann, E., 2012. How Well Are We Measuring Snow: The NOAA/FAA/NCAR Winter Precipitation Test Bed. Bulletin of the American Meteorological Society, 93(6): 811-829.
Rittger, K., Painter, T.H. and Dozier, J., 2013. Assessment of methods for mapping snow cover from MODIS. Advances in Water Resources, 51(1): 367-380.

---

## Author Comment (AC3) · 10 Jan 2018

We thank Referee #3 for his or her constructive comments. We will respond in-line to the critiques.

Review of Bair et al.
General comments
The paper summarizes data, data quality, and data cleaning and interpolation methods from sites operated by the University of California Santa Barbara on Mammoth Mountain, CA. Data from 2011 through 2017 is presented including a continuation of snow energy balance component measurements at the long-running "CUES" site. Coupled with newer intensive measurements of snow depth and density, in particular daily new snow depth and SWE hand measurements, make this a unique and valuable dataset. Figure 8 and associated discussion are particularly valuable contributions and it would benefit from a thorough editing to trim unnecessarily wordy sections and clarify meaning. Examples are mentioned in the technical correction section.
Further, I found the organization of the Datasets section confusing. The first section is on energy balance components, while the second section focused on data filtering and processing. It would be better if the organization were by measurement type (energy balance components, and snow measurements) with methods for data processing and cleaning included under these general headings. Specific headings for each measurement type would also help guide the reader (wind speed / direction, temperature, albedo, etc).

Ok, we agree. We will organize by measurement type with child categories on data processing. We will also include specific headings for each measurement.

Specific comments
1) Please justify using climatological averages to gap fill temperature, RH, and air pressure data. It seems that this would create steps in the data and impose periods of average conditions during periods when the climate was likely not average (and hence went off-line). Why not use linear regression with other sites in the area for gap fill periods greater than 12 hours?

Ok, this is a good suggestion and relatively straightforward for RH and air temperature. Air pressure will likely have to come from the KMMH airport, which is much lower in elevation, thererfore it will have to be lapsed. We will implement these changes.

2) The CUES site is on the north aspect of Mammoth Mountain. Does this topography impact the measurement of direct and diffuse radiation? If so, what is the timing and magnitude of the impact relative to a site without substantial topographic shading?

Not really, except maybe at the highest solar zenith angles. Although the aspect of CUES is north, it is on a nearly flat slope (4°) with a 0.95 sky view factor, computed using a 0.3 m DEM. In terms of shading, as mentioned on p. 7, shadows from vegetation that affect the downlooking radiometers are a more substantial issue than topographic shading.

3) How was the WS600 data used to fill gaps in RM Young 5105 time series (regression, replacement, other)?

Good point. Given that the anemometers are nearly collocated, but at different heights with the 5103 being a few m lower (though still > 6m off the bare ground), we used simple replacement. We will clarify this.

Technical corrections/comments:
Minor comment: Figure 3 would be more effective if it were side-by-side with a snow-off photo and/or with arrows pointing to the instrumentation referenced in the caption.

That's a good suggestion. Please see Bair et al. (2015) for the photo you describe. Unfortunately, we don't have don't have a snow off photo from the exact same vantage point on hand, but the photo in Bair et al. (2015) is from a drought year, showing some of the annual variation in snow depth.

Section 1 Introduction. I suggest replacing "tedious and nontrivial adjustments that are only possible by those intimately familiar . . ." be replaced with something like "sub- stantial and nontrivial adjustments that require detailed information on measurement location characteristics". This is an important subject for all folks in the business of collecting environmental data.

Ok, will do.

Section 2.1 and 2.2. Each section would benefit from an introductory paragraph de- scribing the measurements at the site included in the dataset. It is difficult to sort through what is and is not in the dataset. Separating them would help.

Ok, will do.

Section 3.2.2 Albedo calculation. This is an important section that would benefit from some reorganization, specifically an introduction paragraph that outlines the overall process (e.g. "To calculate albedo, we did 1,2,3,4,5 etc").

Ok, we will create a separate section for the snow albedo calculations and provide an introductory paragraph.

Section 4.1 Moving the description of the data and mention of Figure 4 to the first couple of sentences would greatly clarify this section. Subsequent description of other years outside of the dataset then are placed in context.

Ok, will do.

Bair, E.H., Dozier, J., Davis, R.E., Colee, M.T. and Claffey, K.J., 2015. CUES – A study site for measuring snowpack energy balance in the Sierra Nevada. Frontiers in Earth Science, 3: 58.